# MetaQA: Enhancing human-centered data search using Generative Pre-trained Transformer (GPT) language model and artificial intelligence

Diya Li[1], Zhe Zhang[1,2] *

**1** Department of Geography, Texas A&M University, College Station, Texas, United States of America,
**2** Department of Electrical and Computer Engineering, Texas A&M University, College Station, Texas, United States of America

* zhezhang@tamu.edu

## Abstract

Accessing and utilizing geospatial data from various sources is essential for developing scientific research to address complex scientific and societal challenges that require interdisciplinary knowledge. The traditional keyword-based geosearch approach is insufficient due to the uncertainty inherent within spatial information and how it is presented in the data-sharing platform. For instance, the Gulf of Mexico Coastal Ocean Observing System (GCOOS) data search platform stores geoinformation and metadata in a complex tabular. Users can search for data by entering keywords or selecting data from a drop-down manual from the user interface. However, the search results provide limited information about the data product, where detailed descriptions, potential use, and relationship with other data products are still missing. Language models (LMs) have demonstrated great potential in tasks like question answering, sentiment analysis, text classification, and machine translation. However, they struggle when dealing with metadata represented in tabular format. To overcome these challenges, we developed Meta Question Answering System (MetaQA), a novel spatial data search model. MetaQA integrates end-to-end AI models with a generative pre-trained transformer (GPT) to enhance geosearch services. Using GCOOS metadata as a case study, we tested the effectiveness of MetaQA. The results revealed that MetaQA outperforms state-of-the-art question-answering models in handling tabular metadata, underlining its potential for user-inspired geosearch services.

## 1. Introduction

Addressing societal challenges such as disaster management necessitates the efficient and accurate sharing and usage of geospatial data from various sources. A key part of this process is service matching, where user requirements for spatial data are used to search across archives. However, traditional search approaches, such as the one used by the Gulf of Mexico Coastal Ocean Observing System (GCOOS) [1], part of the U.S. Integrated Ocean Observing System

**Data Availability Statement:** We have uploaded our data and code to the journal recommended repository: https://www.kaggle.com/datasets/dylanli/metadata-question-answering-dataset.

**Funding:** This project is supported by NSF Convergence Accelerator Track E: Combining high-resolution climate simulations with ocean biogeochemistry, fisheries and decision-making models to improve sustainable fisheries (Award Number:2137684). Initials of the authors who received each award: Z.Z, D, L Funder Website: https://www.nsf.gov/awardsearch/showAward? AWD_ID=2137684&HistoricalAwards=false The funders had no role in study design, data collection and analysis, decision to publish, or preparation of the manuscript.

(https://ioos.noaa.gov/), often fall short. These systems, which rely on keyword inputs from users, face limitations due to the inherent uncertainty within spatial information and its representation during the data search process. Furthermore, GCOOS, despite collecting and ensuring the reliability of thousands of data points, presents metadata in complex tables that are difficult to interpret. The system allows data search using keywords or by drawing a polygon on a map, but the returned results, primarily metadata tables, lack detailed descriptions of the data, its usage, and potential linkage with other data products. This limitation makes the system more suitable for experienced researchers familiar with the data and related products, while new users must resort to additional resources to understand the data product and its application in research, a process that can be time-consuming.

The current geo-research system is limited since GCOOS stores the metadata in complex data tables, which is difficult to interpret. As search results, the system returns a metadata table that describes the data title, attribute names, and a data download link. The information about detailed description of the data, use of the data, and possible linkage with other data products is missing from the current GCOOS data-sharing platform. Fig 1 illustrates an example of the GCOOS data search user interface, where users can enter a keyword such as "wind speed" in the keyword field for searching data. After that, the system will return a data table describing the data name and ID, metadata, and download link. The GCOOS advanced data search portal enables users to search for data at a spatiotemporal scale, where users can draw a polygon on the map and specify the data coverage time period. However, the results of the data search remain at the metadata tabular level, which cannot be used to explain more detailed knowledge about the data or any other data or literature that are related to the searched data item. In other words, the current GCOOS data search system is useful for experienced researchers (or users) who know data and related data products well. New users who are unfamiliar with the research area must use other literature or resources to learn the data product and how to use the data in the research, which is time-consuming [2, 3].

Above mentioned research challenges raise a research question: How to develop a user-inspired data-sharing system to support geosearch and make spatial data FAIR principle (Findable, Accessible, Interoperable, and Reusable) [4]? Language models (LMs) have proven to be highly effective in developing Question and Answering systems, sentiment analysis [5], text classification [6], and machine translation [7]. However, despite their success in comprehending free-form natural language sentences by learning from massive amounts of unstructured text data, LMs still face significant challenges when the metadata are represented as tabular question answering. The structure of tables and the relationships between the data are often implicit and difficult to be recognized using raw language models. [8–10]. To address this challenge, various methods have been developed to improve LMs' ability to handle structured data, such as tables. For example, fine-tuning techniques can be highly effective in adapting pre-trained models to specific tasks, and it's promising to see these techniques being applied to absorb table-specific representations [11]. Similarly, the use of synthetic SQL (Structured Query Language) refinement through the generation of artificial queries and tables can help language models learn to better understand and work with structured data [12–14]. These methods are crucial for fabricating user-driven data-sharing frameworks that endorse geosearch and render spatial data FAIR by enabling productive and accurate data exchange [12, 15]. There are various research gaps in the development of tabular question-answering models, including the need for an extensive training corpus of high quality, the requirement to infer correlations among columns and rows, and the necessity to understand the implications of complex structures in the table. Addressing these gaps can result in improved models that are better equipped to reason about the components of the table and provide accurate responses.

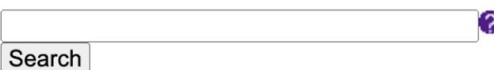

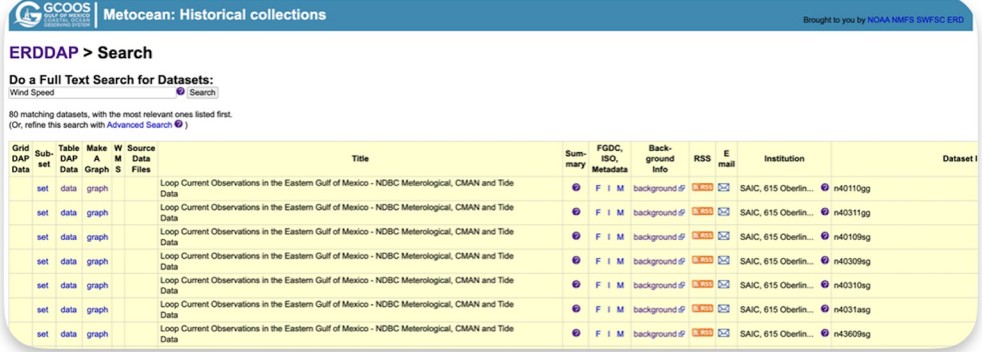

**Fig 1. An illustration of using a keyword-based search in the GCOOS platform.**

In this project, we developed a novel data search model called MetaQA (**Meta**data **Q**uestion **A**nasering) to enable user-friendly geosearch and service matching. The MetaQA system uses Generative Pre-trained Transformer (GPT) Language Model and machine learning algorithms to understand and interpret the user's query and provides relevant answers based on the meta-data and GPT knowledge base. We applied our MetaQA model to GCOOS metadata to help GCOOS improve the data services. Finally, we tested the performance of MetaQA using the Metadata Question Answering data sets by comparing different tabular QA models and showed our MetaQA performed better than work described in [15–17]. Our approach lever-ages the power of the GPT langue model to provide users with rich information about the data and its relevant literature based on the keywords question [18]. Furthermore, the output of the MetaQA system can provide context and background knowledge about the scientific data, which enables researchers to find the relevant data more quickly, ultimately leading to new sci-entific discoveries and insights [18].

The remainder of this article is organized as follows. Section 2 introduces the literature back-ground of our research. Section 3 presents the data collection and processing methods. Section 4 illustrates the MetaQA methodology of the metadata tabular question-answering model. Sec-tion 5 implements experiments by the proposed method, analyzes the results, and discusses this method for different applications and limitations. The last section concludes the study.

## 2. Background

Data-sharing platforms (e.g., GCOOS, National Aeronautics and Space Administration, and U.S. Geological Survey data portals) are becoming increasingly important for researchers in various fields as they enable researchers to access, reuse, and compare data across disciplinary boundaries [19, 20]. By making data more widely available, these platforms can also accelerate scientific discovery, facilitate collaboration, and help ensure the reproducibility of research findings. For instance, researchers working in disaster management can access disaster-related data from various data-sharing platforms and analyze the data to observe climate patterns and conduct disaster mitigation plans [21–24]. Additionally, a growing amount of spatiotemporal data is being collected and analyzed across various domains, including earth sciences, urban science, transportation, epidemiology, and social sciences [25–28]. Spatiotemporal data is dis-tinct from relational data, as it includes both spatial and temporal attributes with multiple actual measurements [26, 29].

The traditional keyword-based geosearch approach may not be sufficient for searching and discovering spatiotemporal data on data-sharing platforms, particularly when spatial informa-tion is uncertain or when the data is scattered across multiple platforms with unstructured metadata [30]. In such cases, alternative approaches such as semantic search, machine learning, and natural language processing may be more effective in discovering and integrating relevant data. These approaches can help to identify relevant data based on the context and meaning of the search query rather than just relying on exact keyword matches. They can also help to iden-tify and extract spatial information from unstructured metadata, and to integrate data from multiple sources based on their semantic similarity [3]. For example, Elnozahy et al. [31] found that the QA system can help to overcome the barriers posed by limited metadata and domain-specific knowledge. By leveraging large language models trained on relevant knowledge, a question-answering system can provide more effective and efficient access to scientific data, enabling researchers to discover and understand the data they need more easily [11, 32, 33].

Recent pre-trained language models such as BERT [6] and BART [7] have achieved state-of-the-art results on a wide range of natural language processing tasks. While language models have shown remarkable success in interpreting unstructured natural language, they can

encounter difficulties when dealing with tabular question-answering tasks where the metadata is represented as structured data [8–10]. Tabular data such as GCOOS data has a different format and structure compared to natural language text. Tabular data is typically represented in a structured format with rows and columns, and each column has a specific data type and format [34, 35]. Different techniques and approaches have been proposed to address this challenge in handling tabular data [17], semantic parsing [36], and entity linking [35]. These techniques aim to extract the relevant information from the table and map it to a structured representation that can be used by language models for question-answering. Semantic parsing techniques [36], on the other hand, involve translating the table schema and contents into a formal representation that can be used by language models for question-answering. The arrangement of tables and the associations between data are often implicit and not easily identified by raw language models. They may fail to deduce the relationships between columns and rows or to grasp the implications of absent values in the table [34, 35]. This shortcoming can lead to incorrect or inadequate answers when confronted with questions that necessitate reasoning about the relationships between various components of the table [5].

Developing tabular question-answering models can be challenging due to the need for a large-scale training corpus with high quality. Tabular data is often complex and heterogeneous, and the natural language queries that are used to retrieve information from the tables can be diverse and nuanced. [15] illustrated challenges, including collecting parallel data comprising natural language sentences and tables as the pre-training corpus, and either crawling tables or synthesizing natural language sentences on available tables. While the crawling approach offers the advantage of a larger corpus, it is also more prone to noise and requires complicated heuristics to clean [15]. On the other hand, the synthesis approach offers better control over the quality of the data, but it is also more costly and often lacks diversity. Recent studies [37, 38] found that the use of the generative pre-trained transformer (GPT) model to build a training corpus can accelerate the process of synthetic data, which allows researchers to obtain a larger and more diverse corpus without relying on manual annotation or data crawling [37]. Moreover, researchers can fine-tune GPT on a smaller annotated corpus (e.g., metadata tables) and questions, allowing the model to learn the specific patterns and relationships in the data that are relevant to the tabular question answering task [38].

Recent research [39] on prompt tuning in NLP has shown good performance in downstream tasks. Compared to traditional fine-tuning methods, prompt-tuning can use a smaller dataset consisting of carefully crafted prompts to optimize the performance of a pre-trained model on specific tasks [39]. The goal of prompt learning is to automatically generate a diverse set of high-quality questions that represent the data and can be used to train and evaluate question-answering models [40]. Similar to synthetic fine-tuning [15], using a template to generate and fit the parameters during training makes the model easier to understand and learn the patterns and relationships in the data. While the large language model and prompt tuning method show promise for generating high-quality questions from metadata tables, more research is needed to explore their potential for supporting metadata search and discovery [41, 42]. Prompt tuning typically follows the design of verbalizer mapping [43]. For example, if the input format is a structured query that requires a specific output format, such as a table or a list, the verbalizer mapping would convert the output of the language model into the appropriate format [44]. Similarly, if the input format is a natural language question, the verbalizer mapping would convert the output of the language model into a grammatically correct and meaningful sentence that answers the question [43]. Designing the verbalizer mapping in prompt tuning requires a lot of effort to test the effectiveness of the template [39]. Existing

research has found that using a much larger general language model, such as GPT-3.5-turbo, can handle template generation [45].

## 3. Data

In this project, we used two types of data sets: The GCCOS metadata and the Wikipedia Tabular Question Answering Dataset. The GCOOS data was used to create a metadata question answering dataset and the Wikipedia Tabular Question Answering Dataset was used to fine-tune the model to understand free-form tabular question answering.

As shown in Table 1, we evaluated the performance of our MetaQA model using Wikipedia Tabular Question Answering (WikiTQA) Dataset [36] and Metadata QA datasets. The GCOOS metadata is downloaded from the Gulf of Mexico Coastal Ocean Observing System (GCOOS) data service (https://gcoos.org/). Three kinds of data products were collected from the ERDDAP data servers (https://data.gcoos.org/erddap.php)including the historical collection of oceanographic and meteorological data, near-real-time observation and biological & socioeconomics data. In this project, we built a metadata table question-answering dataset by collecting 25,810 raw metadata from GCOOS. The attributes used for building the Metadata Question Answering dataset are described in Table 2. After preprocessing the raw metadata, 890 metadata tables were selected. The following data preprocessing steps were included:

- Data Cleaning: Removing duplicate, missing, or irrelevant data to ensure the quality of the dataset.

- Data Normalization: Converting data into a standard format, such as converting text to lowercase or converting dates into a standard format.

- Data Integration: Merging data from different sources to create a unified dataset.

- Data Transformation: Transforming the data into a suitable format for use in the question-answering system, such as converting text into numerical values or creating a table structure from unstructured data.

- Data Sampling: Reducing the size of the training batch by sampling only a batch of rows and columns to make sure the input data satisfies the maximum input tokens.

Using the prompting method, over the example question-answer pairs were generated from these tables. This dataset will serve as a valuable resource for developing and evaluating AI models for question-answering systems, particularly for those that operate on metadata tables.

**Table 1. Dataset statistics.**

|  | WikiTQA | Metadata Question Answering dataset |
|---|---|---|
| QA Examples | 22033 | 11926 |
| Tables | 2108 | 1,433 |
| Complex QA [1] | yes | yes |
| Metadata | n/a | 24,788 |
| Spatial-temporal | no | yes |

Complex question answering refers to a complex and multi-step reasoning process to arrive at the answer (e.g. How. . .?, What is the most. . .? ).

**Table 2. The example attributes in the metadata tables.**

| Attributes | Examples |
|---|---|
| feature type | Trajectory |
| geospatial bounds coordinate reference systems | EPSG: 4326 |
| date created | 2019–10–16T12:35:42Z |
| geospatial latitude maximum | 29.34 |
| geospatial latitude minimum | 27.87 |
| geospatial latitude units | degrees north |
| geospatial longitude maximum | -93.29 |
| geospatial longitude minimum | -94.81 |
| geospatial longitude units | degrees east |
| summary | The main focus of this cruise was to monitor the abundance and size of lionfish and remove as many as possible of this invasive species. . . |
| time coverage start | 2015–06–08T21:34:00Z |
| time coverage end | 2015–06–11T23:26:21Z |
| data product title | RV Manta—FGBNMS—FGBNMS-15–03_Lionfish_Removal_1.nc |

## 4. Methdology

### 4.1 Masked language model

Masked language models involve predicting a masked word or phrase within a sentence or sequence of text [6]. By replacing certain tokens or words with a special token, often denoted as [MASK], these models are trained to predict the original word or token based on the context of the surrounding words. In this project, we applied a modified Bidirectional and Auto-Regressive Transformer (BART) [7] as the base language model as our backbone. It is a large-scale sequence-to-sequence model that uses a combination of both autoencoder and autoregressive architectures for generating text. For example in Fig 2, given the sentence "New York City is [MASK] region", the weight of the masked language model can learn by predicting the process of the missing word "east" or "south" based on the context of surrounding words. This training process will let the model learn to recognize the underlying structure of the text and to be more robust to noise or errors in the input. This approach can be applied to question-answering tasks by using the model to predict the answer to a given question based on a given context. In this project, we modified the BART-large model that follows a standard sequence-to-sequence Transformer architecture, with GeLU activation functions for a better smooth effect expressed as below:

$$\text{GeLU}(x) = 0.5 \cdot x \left[ 1 + \text{erf}\left( \frac{x}{\sqrt{x}} \right) \right], \tag{1}$$

where erf is the error function used in statistics to calculate the probability of an event. Adopting the question-answering task to the masked language model is similar to the text completion example shown above. The input of the model that is passed to the encoder contains a natural language sentence along with a corresponding table. It is crucial to mention that the quality of this natural language sentence significantly impacts the quality of the output. The sentence, ideally a question, should be clear, concise, and directly related to the data in the table for the model to provide a precise and accurate answer. Ambiguous or overly complex questions might lead to inaccurate or vague answers. Furthermore, both the natural language sentence and the table should be stringified so they can be transformed through tokenization,

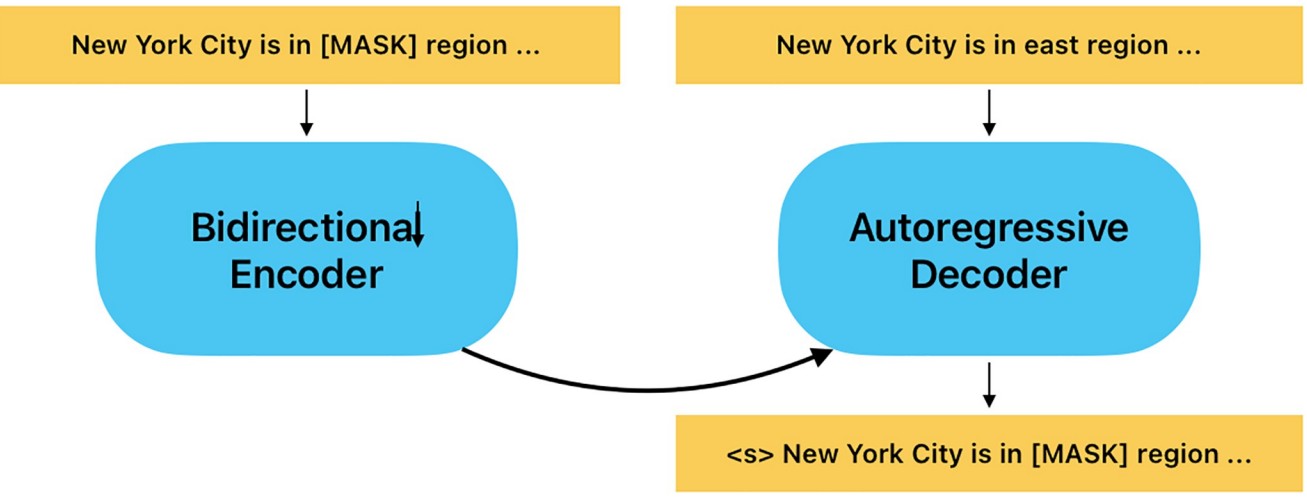

**Fig 2. The demonstration of the BART model: A document with random errors or noise (left) is considered "corrupted".** The model encodes this corrupted document bidirectionally and then uses an autoregressive decoder to calculate the likelihood of the original, error-free document (right). To fine-tune the model, an uncorrupted document is inputted to both the encoder and decoder [7].

which breaks down the text into smaller units. Tokenization is an essential step in NLP task as it allows the text to be processed by a computer in a more structured and efficient manner.

### 4.2 Metadata tabular question-answering

In MetaQA, we employed an encoder-decoder BART model and applied pre-training as a sequence generation task for Pre-trained LM from free-form NL, as shown in Fig 3 as starting

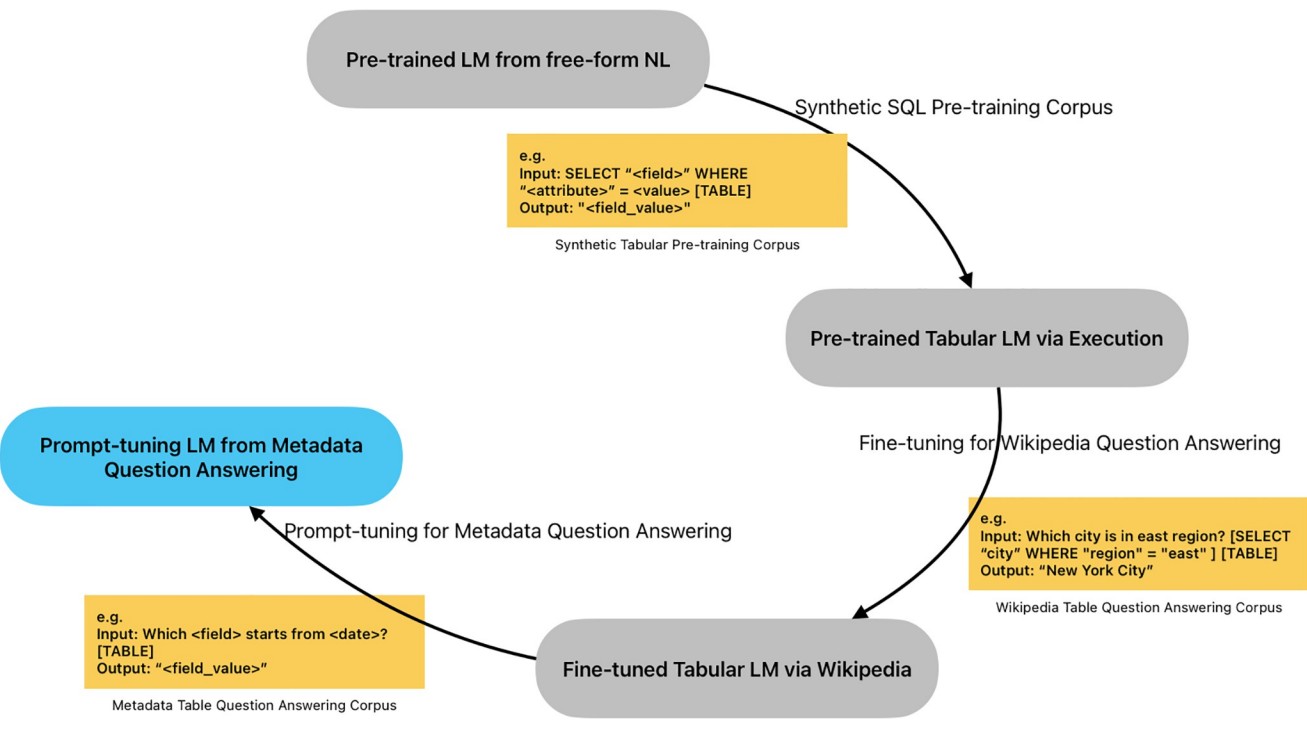

**Fig 3. The overview of MetaQA model.**

point. We followed the table pre-training via execution [15] training architecture that applies tabular question-answering downstream tasks with an end-to-end workflow, which starts with a BART base model as a pre-trained free-form language model to conduct the SQL synthetic. After that, the pre-trained LM is tuned with a multi-round tuning method with various datasets including Wikipedia Question Answering Dataset and Metadata Question Answering Dataset.

**4.2.1 Spatial-temporal SQL synthesizes.** Many geoscience data such as GCOOS data contain spatial-temporal information, which is difficult to retrieve during the data search process [46]. To design a suitable architecture for metadata tabular question-answering, we argue that the structured spatial-temporal information in metadata can be applied to the language model via SQL command, as the traditional search method often use it to find out the exact result by SQL execution in a relational database (e.g. MySQL). However, the existing SQL synthesizing method lacks spatial-temporal relationship [15]. For example, as shown in Fig 4, we adopt the spatial-temporal SQL command conducted during the synthetic step. The synthesized dataset is automatically generated and sampled from `SQUALL` dataset [14] with the spatial-temporal keywords insertion. An example SQL template is `SELECT data_product WHERE start_date >= val`, where `data_product` and `start_data` can be denoted as the columns name with the temporal attributes and value respectively and `val` represents the value stored in the certain rows. Note that the spatial-temporal query is a complex operation in Geographic Information Science (GIS). By tracing back to the base operation in the SQL execution level, we simplified the operations into eight categories, those are: select, filter, aggregate, superlative, arithmetic, comparative, group, and spatial union&intersection. In this step, the input text is the stringified SQL command as well as the corresponding tables. In practice, the table is flattened as a long-form text so that it can be passed directly into the tokenizer. Before flattening the table, the table was pre-processed with various special indicators to address the table structure in a natural language way. For example, a table $T$ with $n * m$ cell structure can be flattened as

$$T = [\text{HEAD}], c_1, ..., c_n, [\text{ROW}], 1, r_1, [\text{ROW}], 2, r_2, ..., r_m[\text{TAIL}], \qquad (2)$$

where [HEAD], [ROW] and [TAIL] are indicators to locate the headers, rows, and the end of the table. This way, the spatial-temporal attributes are addressed during the training steps by applying multiple spatial-temporal queries.

**4.2.2 Prior knowledge fine-tuning.** Prior knowledge fine-tuning is an important step in metadata tabular QA because it allows the model to learn how to effectively use external information to improve its performance on the task [47]. The previous pre-trained language model is trained on large amounts of text data without targeting scientific question-answering tasks [48]. By fine-tuning a pre-trained language model on a spatial data metadata dataset, the model can learn to make more accurate predictions without modifying the architecture or parameter dimension of the model. WikiTQA dataset [36] is a scientific-related question-answering dataset designed to test the performance of the table question-answering model in the downstream task. By manually labeling the question-answering set on thousands of Wikipedia tables, the LM fine-tuned on WikiTQA can take free-form natural language questions, instead of SQL commands, to answer the question by simply inferencing their fine-tuned weights. In this step, the pre-trained model is further trained with a smaller dataset and similar input and output. The detailed fine-tuned steps are:

1. **Select a pre-trained LM**: By loading the pre-trained BART model with 400 million parameters on SQL synthesizes, we initialize the fine-tuning model weight appropriately.

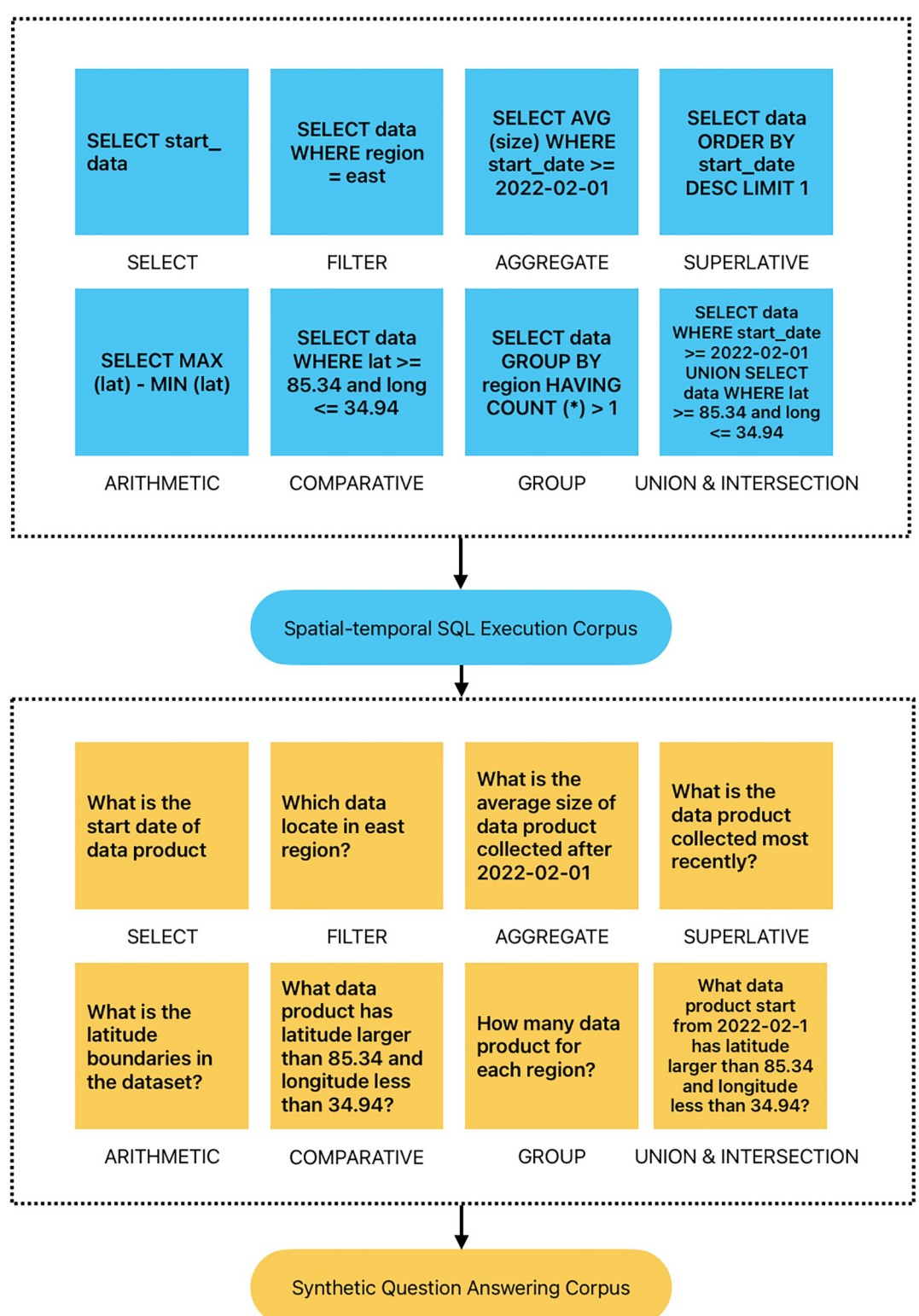

**Fig 4. The demonstration of transform from executable SQL command to question.**

2. **Prepare the data**: We collect and preprocess the dataset in this step. The preprocessing step is mentioned in Section 3.

3. **Fine-tune the model**: In this step, we fine-tune the model with *Adam* optimization algorithm [49] by updating the model's parameters to minimize a cross-entropy loss function on the *fairseq-lib* [50] that measures the difference between the model's predictions and the true labels.

4. **Evaluate the model**: After training, evaluate the model's performance on validation and testing sets. The evaluation metric is denotation accuracy, which is described in Section 5.

**4.2.3 Question-answering prompting.** Prompt tuning, the process of fine-tuning a pre-trained language model (LM) on a specific domain using relevant prompts or examples, has been widely recognized for its efficacy in domain-specific tasks [40]. The literature proposes various strategies to enhance the performance of LMs in specific domains through prompt tuning. In our approach, we forgo the use of complex templates for mapping predictions from fine-tuned models to verbalizers [51]. Instead, we harness the capabilities of a more powerful general language model, specifically an Artificial General Intelligence (AGI), to backtrack original questions from the provided answer template. We assert that the answers generated from the tabular question-answering system should have a strong correlation with the information of specific cells. As illustrated in Fig 5, we employ `gpt-3.5-turbo` [52] via LangChain (https://github.com/hwchase17/langchain) to generate question-answering templates. These templates incorporate constraints such as question type, device location, temporal features, or spatial features. In this system, we propose integrating a two-step chain-of-thought prompting (CoT) design into our metadata question-answering system. CoT prompting, a method that has shown substantial potential in enhancing the reasoning capabilities of large language models, is particularly effective in tasks requiring complex, multi-hop reasoning [53]. In our

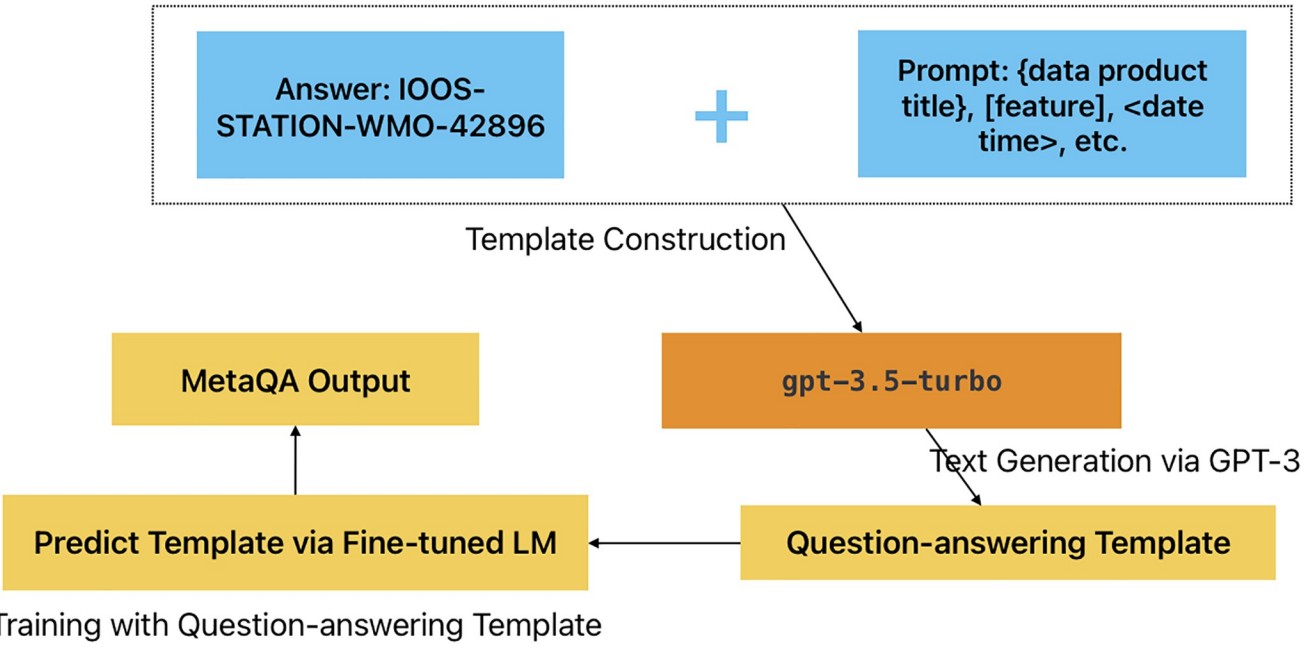

**Fig 5. The demonstration of transform from executable SQL command to question.**

**Table 3. The example questions and answers for our model on experimental datasets.**

| no. | Example Questions | Example Answers |
|---|---|---|
| 1 | What is the name of the data product that contains current profile data from a Downward-looking Teledyne RDI 38 kHz Ocean Observer ADCP in Sevan Louisiana from 2017/04/25 to 2017/06/06? | Data for ioos-station-wmo-42896 |
| 2 | What data product is used to study the linkage between known sandy deposits on the shelf edge and deepwater Rio Grande Fan? | RV Manta—FGBNMS—FGBNMS-15–27_UTIG_2_1.nc |
| 3 | Where was FGBNMS-15–27 collected? | 29.31N, 94.82W. |
| 4 | What data product was used to measure during Tropical Storm Bill in June 2015? | RV Manta—FGBNMS—FGBNMS-15–18-Tropical_Storm_Bill_Jun_2015_1.nc |
| 5 | What is the purpose of FGBNMS-15–09? | FGBNMS-15–09 is to continue the long term monitoring effort at Stetson Bank, including photographing long-term photo-stations and conducting randomized transects and fish surveys. |
| 6 | What data product is used to initialize long term monitoring studies of mesophotic coral ecosystems? | RV Manta—FGBNMS—FGBNMS-15–12_ROV-Stetson_Bank_1.nc |
| 7 | What was the cruise designator for FGBNMS-17–06? | The cruise designator for FGBNMS-17–06 was WTX01. |

context, the CoT design involves planning the analysis goal and identifying relevant metadata terminology. This process provides a sequence of thought demonstrations or 'chains' that guide the language model in its reasoning process. Each 'thought' corresponds to a specific piece of metadata, and the model sequentially processes these 'thoughts', effectively decomposing the complex reasoning task into manageable steps. For instance, given the first example in Table 3, the first CoT is designed to identify the data product id (e.g., "IOOS-STATION-WMO-42896") based on the sampled cell information from the metadata tabular from a miscellaneous table content. The second step involves applying the generated indicator field (e.g., "{data product) to the LLM chain to support the previously fine-tuned question-answering system. These pre-defined LLM chains can be applied iteratively with a looping condition to enhance the quality of the question with a set of temperatures, also known as the diversity parameter.

## 5. Experiments and results

### 5.1 Experiments

The implementation of this project was carried out on the FASTER cluster, a high-performance computing resource provided by Texas A&M University and funded by the National Science Foundation. The training process utilized the NVIDIA A100 GPUs and lasted approximately 40 hours, encompassing up to 20,000 steps. A batch size of 512 was employed, accompanied by an initial learning rate of 3e-5, which decayed polynomially. Given the substantial amount of metadata involved, a maximum tokenization length of 4,000 was defined, with any exceeding tokenized text being truncated. During the training phase, a warm-up stage was incorporated to assist the model in avoiding suboptimal solutions early in the process. This warm-up stage allowed for gradual increases in the learning rate, enabling the model to explore the solution space more effectively and achieve better convergence within a shorter time frame. In our experiments, the warm-up stage persisted for 5,000 steps. To mitigate the frequency of parameter updates, gradient accumulation was employed every 12 epochs. This approach reduced the computational burden by reducing the number of updates made to the model's parameters. Furthermore, evaluation metrics were computed every 1000 epochs using the equation specified in the subsequent section. The evaluation procedures were conducted through ten independent experiments, ensuring the robustness and reliability of the results.

The fixed random seeds list ranging from 42 to 52 was utilized, enhancing the reproducibility of the findings. The average evaluation derived from these experiments is presented later in this section, providing a comprehensive summary of the overall performance.

## 5.2 Evaluation

**5.2.1 Denotation accuracy.** Denotation accuracy in question-answering language models refers to the ability of the model to correctly identify and output the correct answer to a given question. This accuracy is typically measured by comparing the model's output to the answer and calculating the proportion of questions for which the model's output is exactly the same as the answer:

$$acc = \frac{c}{Q}, \tag{3}$$

where $c$ denote as number of correct answers and $Q$ denote as total number of questions. As shown in Table 4, the denotation accuracy is being compared with a recent state-of-the-art table question-answering model trained using fine-tuning with prior knowledgeThe true answers are sourced from the WikiTQA benchmark dataset.

**5.2.2 Soft denotation accuracy.** Metadata table question-answering usually generates a more complex answer, which contains a detailed date-time and latitude, and longitude. For example, the answer "29.31N, 94.82W" will be considered as a wrong answer compared with the correct one "94.82W, 29.31N". [16] found that a soft form of accuracy measurement can get an overall accuracy to evaluate the performance of key information retrieving ability of the question-answering model. As it relaxes the strict equal formula to a task-specific similarity score:

$$acc(x, \hat{x}) = \begin{cases} 1 \text{ if } x = \hat{x} \\ 0 \text{ if } x \text{ is off} - \text{topic (e.g. n/a)} \\ p \, p = \frac{1}{|\hat{x}|} \sum_{\hat{x}_j \in \hat{x}} max_{x_i \in x} \mathbf{x}_i^{\mathsf{T}} \hat{\mathbf{x}}_j \end{cases}, \tag{4}$$

where $p$ denote a precision score, $x$ denotes a predict text and $\hat{x}$ denotes reference text from dataset. The precision score is commonly used in evaluating the text generation task [16, 54] by matching each token in x to a token in $\hat{x}$ to a token in $x$ to compute maximum precision with the greedy method, where each token is matched to the most similar token in the other sentences. Similarly, the recall score that gives us the percentage of positives well predicted by our model can be expressed as:

$$recall(x, \hat{x}) = \frac{1}{|x|} \sum_{x_i \in x} \max_{\hat{x}_j \in \hat{x}} \mathbf{x}_i^{\mathsf{T}} \hat{\mathbf{x}}_j, \tag{5}$$

and the F1 score that measures both precision and recall by the harmonic mean of the two is

**Table 4. Denotation accuracy.**

| Model | Accuracy |
|---|---|
| Tapas [16] | 0.488 |
| Omnitab [17] | 0.613 |
| Tapex [15] | 0.575 |
| **MetaQA** | **0.598** |

expressed as follows:

$$F(x, \hat{x}) = 2 \frac{acc(x, \hat{x}) \cdot recall(x, \hat{x})}{acc(x, \hat{x}) + recall(x, \hat{x})}.$$  (6)

As shown in Fig 6a and 6b, the two tokenized sentences are compared according to the accuracy score. The overall accuracy in Fig 6a and 6b is 0.9879 and 1.000 respectively.

## 5.3 Results and discussion

**5.3.1 Evaluation metrics comparison.**   Tables 5 and 6 indicates that our fine-tuned model outperforms several similar existing models [11, 16, 17] in metadata question answering on the Metadata Question Answering datasets. Specifically, our model achieved an F1 score of 0.8911, a precision score of 0.9034, and a recall score of 0.8891, which surpasses the baseline of other methods. The evaluation step was carried out on an untrained test dataset that was randomly sampled from the original Metadata Question Answering dataset. Moreover, we also included our experiments in Wikipedia Table Question Answering Datasets.

**5.3.2 Example results.**   Table 3 shows an example of our model's output in a standard free-form natural language format, which enables users to ask questions related to desired metadata table and receive relevant answers. For instance, in the first example, the user expects to retrieve the name of a data product that contains technical specifications. Our model retrieves the information presented in the '*summary*' columns and returns the name of the data product from the '*data product title*' column. In the third example, the user expects a result with location information. Our model organizes the information found from '*geospatial latitude maximum*', '*geospatial longitude maximum*', '*geospatial latitude minimum*', etc., to provide a brief latitude and longitude answer. The fourth example showcases our model's ability to find answers from metadata pertaining to a specific storm event, as long as the relevant records are present in the metadata of the data center.

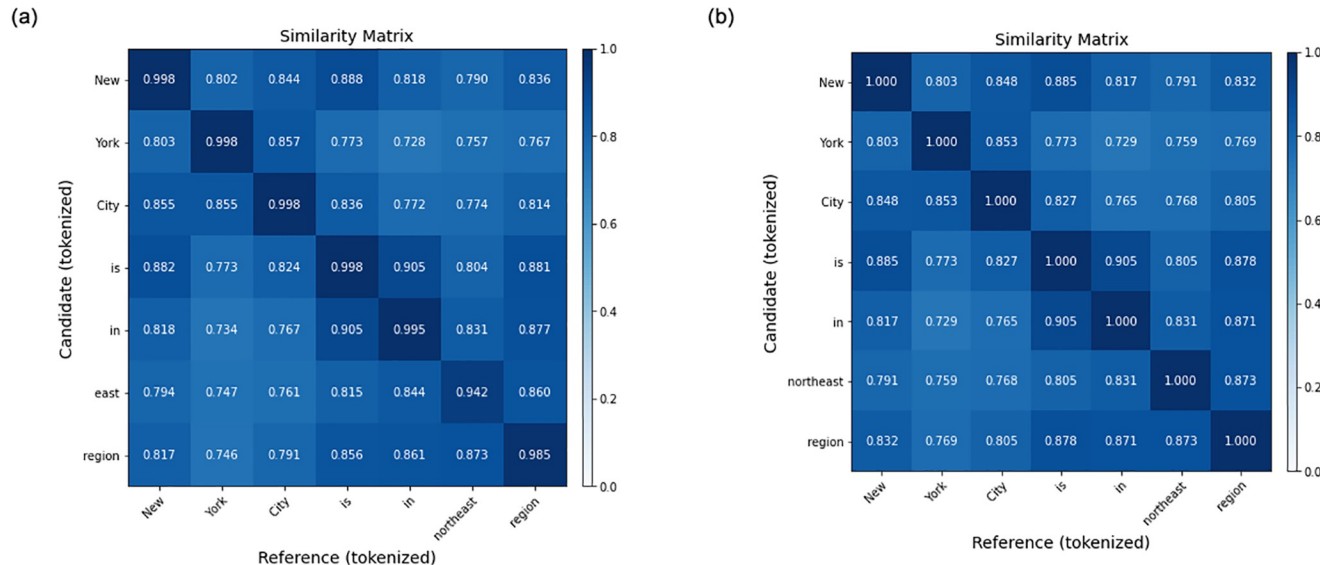

**Fig 6.  The demonstration of comparing the accuracy between the predicted candidate and the actual reference.** (a) The comparison between "New York City is in east region" and "New York City is in northeast region". (b) The comparison between "New York City is in northeast region" and "New York City is in northeast region".

**Table 5. Average soft accuracy on metadata question answering datasets.**

| Model | F1 | Accuracy | Recall |
|---|---|---|---|
| Tapas [16] | 0.8231 | 0.8194 | 0.8464 |
| Omnitab [17] | 0.8631 | 0.8796 | 0.8631 |
| Tapex [15] | 0.9198 | 0.8625 | 0.7820 |
| **MetaQA** | **0.8911** | **0.9034** | **0.8891** |

**Table 6. Average soft accuracy on wikipedia table question answering datasets.**

| Model | F1 | Accuracy | Recall |
|---|---|---|---|
| Tapas [16] | 0.8567 | 0.8284 | 0.8905 |
| Omnitab [17] | 0.9050 | 0.9007 | 0.9106 |
| Tapex [15] | 0.9412 | 0.9359 | 0.9475 |
| **MetaQA** | **0.9087** | **0.8824** | **0.9091** |

**5.3.3 Intuitive comparison.** By sampling a few typical question-answering sets from the Metadata Question Answering dataset, Table 7 provides an intuitive comparison between our model and OmniTab [17], The inference result is directly generated by a stringified metadata table and question, where inference refers to the process of using a trained model to generate predictions or outputs for new input data by passing the input through the model and using its learned parameters to generate a prediction. The comparison results demonstrate that our model is capable of handling spatial-temporal question-answering tasks better than OmniTab, which achieved the second-best performance score in Table 7. This is because OmniTab can only retrieve information directly from the cell, whereas our MetaQA model can extract information and generate natural language answers for abstract questions. While both models perform well when the answer requires only the retrieval of latitude and longitude information directly from cells, MetaQA outperforms OmniTab for complex question settings that require information extraction and natural language generation. Additionally, the first example in Table 6 illustrates that MetaQA can generate a simpler answer format than OmniTab, which only combines the raw text from the '*time coverage start*' and '*time coverage end*' cells.

**5.3.4 Inference efficiency.** In addition to accuracy, the efficiency of the inference process is a crucial metric in the application of large language models. Poor performance in this regard could severely limit the practical usability of the model in production environments. As part of

**Table 7. The comparison between MetaQA and recent table question answering tables.**

| Example Questions | MetaQA | OmniTab [17] |
|---|---|---|
| What is the date range for Data for ioos-station-wmo-42856? | 2012/12/23 to 2014/01/04 | 2012–12–23T23:34:00Z,2014–01-04T23:39:41Z |
| What is the feature type of FGBNMS-15–07_NCRMP1_1.nc? | trajectory | trajectory |
| Where was FGBNMS-15–07_NCRMP1_1.nc collected? | 29.31N, 94.82W | 29.31N, 94.82W |
| What is the purpose of FGBNMS-16–08_SB_LTM_1.nc? | Conduct SCUBA operations at Stetson Bank to continue the long-term monitoring effort. | n/a |
| What is the name of the data product for the Downward-looking Teledyne RDI 38 kHz Ocean Observer ADCP from 28.89N, 87.77W on 2014/01/01 through 2014/04/03? | ioos-station-wmo-42867–201401 | n/a |

(a)

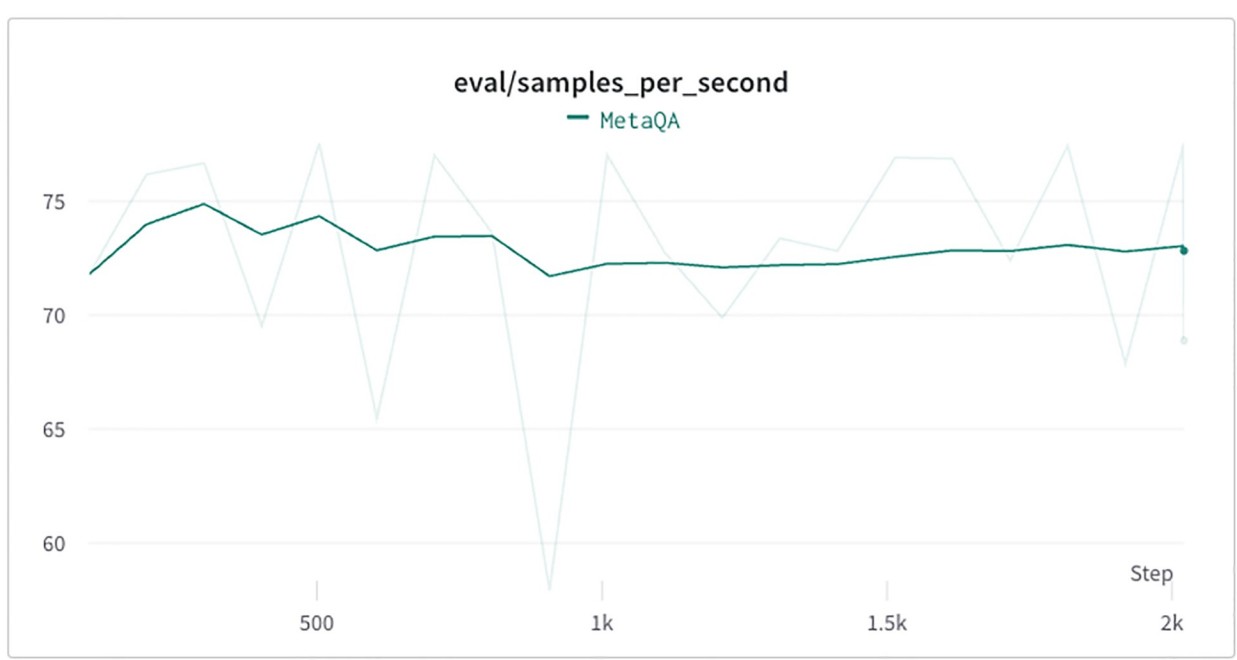

(b)

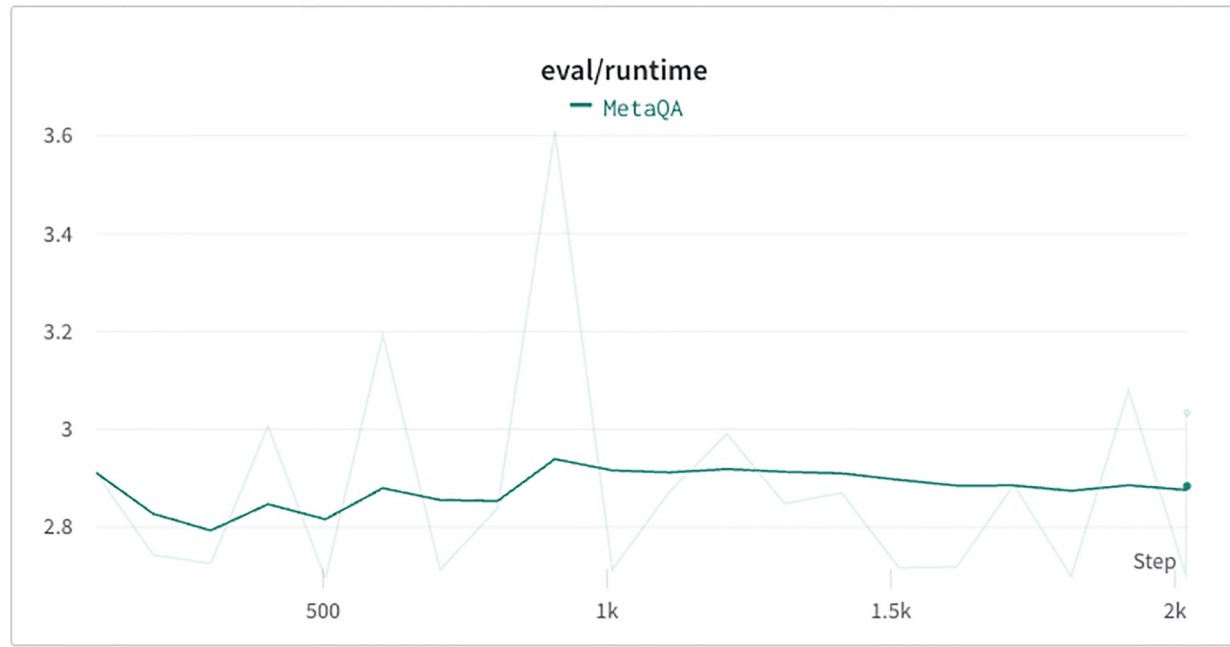

**Fig 7. The efficiency metric recorded during the evaluation steps.** (a) The runtime in second for every evaluation step. (b) The number of processed samples per second for every evaluation step.

our evaluation process, we recorded the inference speed of our developed model every 1000 epochs. Fig 7a shows that the runtime for each evaluation was consistently around 2.88 seconds, with a maximum of 3.609 seconds and a minimum of 2.695 seconds. Meanwhile, Fig 7b reveals that our model was able to process approximately 65 samples per second during the

evaluation steps. This translates to a cost of less than 0.02 seconds for each question-answering task. Overall, our developed model demonstrated high efficiency in inference while maintaining impressive accuracy, making it a practical and reliable choice for various applications.

**5.3.5 Limitations.** One major limitation of our approach is its inability to handle large metadata databases. While our flattening technique works well for relatively small tables, its performance deteriorates as the table size increases. This is due to the fact that our adopted tokenizer has a maximum tokenization length, and even increasing the limit would not enable us to load the entire metadata into memory at once. Another limitation is the specificity of our approach to a particular type of metadata stored in structured databases. The performance and applicability of our method may vary when applied to different datasets (WikiTQA). To address these limitations, we plan to explore compression techniques that can reduce the size of the metadata table without sacrificing any detailed information. Additionally, we aim to develop strategies for handling different types of datasets, including those with unstructured or semi-structured formats. Furthermore, some rows and columns, such as '*geospatial latitude units*', are rarely used and are more akin to scale values. We believe that creating a set of scale values a priori can help reduce the size of the metadata table. In future research, we plan to develop an evaluation method that can balance the influence of adopting AGI models in the end-to-end framework. By doing so, we can ensure that our approach remains reliable and robust for a wide range of applications.

## 6. Conclusion

In this project, we developed a MetaQA model to enable user-friendly spatiotemporal data search and sharing using a large language model. The paper identified the drawbacks of traditional geosearch methods that rely on users' keyword-based search, such as lack of knowledge and descriptions of data, and complex metadata queries (e.g., tabular metadata format). It addressed them by designing an efficient end-to-end QA system. The development of the MetaQA system involves the use of NLP techniques to create a structured and machine-readable representation of datasets and their associated metadata. This can help researchers identify datasets that are most relevant to their research questions and improve the efficiency of the discovery process. The findings of this research demonstrate the effectiveness of the MetaQA model in outperforming other question-answering models, which provides a more efficient and effective way to access and analyze the large amount of data generated by scientific facilities. Overall, the development of this MetaQA system represents an important contribution to the field of GIScience, as it has the potential to improve the efficiency and effectiveness of interdisciplinary research by enabling scientists to discover and access datasets of interest across domain boundaries.

## Supporting information

**S1 Data.**
(BST)

## Author Contributions

**Conceptualization:** Diya Li.

**Data curation:** Diya Li, Zhe Zhang.

**Formal analysis:** Diya Li.

**Funding acquisition:** Zhe Zhang.

**Methodology:** Diya Li, Zhe Zhang.

**Project administration:** Zhe Zhang.

**Resources:** Diya Li.

**Software:** Diya Li.

**Supervision:** Zhe Zhang.

**Validation:** Diya Li.

**Visualization:** Diya Li.

**Writing – original draft:** Diya Li, Zhe Zhang.

**Writing – review & editing:** Diya Li, Zhe Zhang.

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
