## [Decision Letter · Decision Letter 0]

30 May 2023

PONE-D-23-10676MetaQA: Enhancing Human-Centered Data Search using Generative Pre-trained Transformer (GPT) Language Model and Artificial IntelligencePLOS ONE

Dear Dr. Zhang,

Thank you for submitting your manuscript to PLOS ONE. After careful consideration, we feel that it has merit but does not fully meet PLOS ONE’s publication criteria as it currently stands. Therefore, we invite you to submit a revised version of the manuscript that addresses the points raised during the review process.

We look forward to receiving your revised manuscript.

Kind regards,

Baby Gobin

Academic Editor

PLOS ONE

“Funding: This project is supported by NSF Convergence Accelerator Track E: 437

Combining high-resolution climate simulations with ocean biogeochemistry, 438

fisheries and decision-making models to improve sustainable fisheries (Award 439

Number:2137684).”

“This project is supported by NSF Convergence Accelerator Track E: Combining high-resolution climate simulations with ocean biogeochemistry, fisheries and decision-making models to improve sustainable fisheries (Award Number:2137684).

Initials of the authors who received each award: Z.Z, D, L

Funder Website: https://www.nsf.gov/awardsearch/showAward?AWD_ID=2137684&HistoricalAwards=false

Reviewers' comments:

Reviewer's Responses to Questions

**Comments to the Author**

1. Is the manuscript technically sound, and do the data support the conclusions?

Reviewer #1: Yes

Reviewer #2: Partly

2. Has the statistical analysis been performed appropriately and rigorously? 

Reviewer #1: No

Reviewer #2: No

3. Have the authors made all data underlying the findings in their manuscript fully available?

Reviewer #1: No

Reviewer #2: No

4. Is the manuscript presented in an intelligible fashion and written in standard English?

Reviewer #1: Yes

Reviewer #2: Yes

5. Review Comments to the Author

Reviewer #1: MetaQA: Enhancing Human-Centered Data Search using Generative Pre-trained Transformer (GPT) Language Model and Artificial Intelligence

The paper presents a spatial data search model called Meta Question Answering System (MetaQA), which uses a generative pre-trained transformer (GPT) model.

The paper presents a useful application: it proposes a model to build a tabular question-answering system.

As stated in line #234, “The input of the model that passes to the encoder contains a natural language sentence with a corresponding table.” I want to ask the authors about the type of sentences that better fit the problem: what makes a sentence appropriate for this task? The quality of the input sentence influences the quality of the output?

This sentence, “We followed the table pre-training via execution [15] training architecture that applies tabular question-answering downstream tasks.”, is unclear.

Results show that the approach is promising.

General comments:

The resolution of some figures (such as Figure 1) is very poor.

Typos:

- “fromat” – abstract

- “he ..” – Figure 4 caption

Reviewer #2: The authors propose the MetaQA model, which employs machine learning and the GPT Language Model to enhance the searchability and comprehension of geospatial metadata. I found the paper interesting, and its results as impressive.

However, I have a few concerns:

The proposed method is evaluated only on a single dataset. The performance gap between proposed methods and existing methods is impressive, yet the experiment is confined solely to the Metadata Question Answering dataset. Given that the authors themselves created Metadata Question Answering dataset, the robustness of the proposed method needs to be reinforced with other Table QA datasets or similar ones.

Another issue is that the authors did not detail the statistical strength of the experiments. As the authors created the dataset, the statistical strength of the results is critical to establish the credibility of the findings. One example of reporting statistical values could be the average and standard deviation from 10 independent experiments with different random seeds; this could serve as proof of the result. The random seed itself should be also denoted.

Lastly, Metadata Question Answering dataset has not been fully released or made publicly accessible.

The PLOS Data policy denotes that the resources should be fully available without restrictions. According to the same policy, reviewers should have access to the materials and the codes. However, I was unable to access either the dataset or the code. The current statement (Lines 441-443) doesn't fully align with the "available without restriction" policy, which may inhibit the reproducibility of this work. This also makes huddle for review process, as the reviewers cannot contact the authors for the resources due to Confidentiality.

For better reproducibility, I recommend the authors to release the dataset and code in a fully public manner.

Due to the points mentioned above, I am reluctant to decide whether the proposed model is improved model than the other models.

Given that the proposed model demonstrates remarkably high improvement in performance, I believe the paper will have a significant impact if the results and figures are statistically proven. Therefore, both the model and the dataset must be thoroughly inspected and verified before acceptance.

The Introduction and Background sections have redundant content, making them feel excessively lengthy. Conversely, the implementation details are either missing or minimal.

Minor points:

The quality of the figures is low, making it hard to read important information; for instance, the y-axis of the matrix graphs is indistinguishable.

Additionally, the figures are not embedded within the manuscript. I struggled consistently to match the captions with the figures located at the end of the document.

I detected several typos. Here are a few examples:

Line 61: Anaswering -> Answering

Line 69: langue -> language

The authors should thoroughly review the manuscript to rectify any remaining typos throughout the paper.

6. PLOS authors have the option to publish the peer review history of their article (what does this mean?). If published, this will include your full peer review and any attached files.

Reviewer #1: No

Reviewer #2: No

---

## [Author Response · Author response to Decision Letter 0]

7 Sep 2023

Dear Reviewers,

We want to express our sincere gratitude for your valuable feedback and constructive suggestions on our manuscript, "MetaQA: Enhancing Human-Centered Data Search using Generative Pre-trained Transformer (GPT) Language Model and Artificial Intelligence". We have carefully considered your insightful comments and have implemented appropriate revisions to address your concerns. Please find our point-by-point responses to your feedback below.

Reviewer #1:

● Pertaining to the type of sentences that better suit the problem and the influence of input sentence quality on the output:

We appreciate your thoughtful inquiry. In the context of a question-answering task employing a masked language model, the input sentence should ideally be a query that directly pertains to the data represented in the table. It should be articulated clearly and concisely to ensure the model's accurate understanding of the question and subsequently, a precise response. Ambiguities or excessive complexities may detract from the precision of the answer.

Given that the model is essentially a statistical pattern matcher trained on large datasets, it can only provide answers based on patterns it has learned during training. Thus, if the input query does not align closely with these patterns, the model may encounter difficulties in generating high-quality responses.

Consequently, the quality of the input sentence significantly influences the quality of the output as it primes the model for pattern matching during the question-answering process.

We have revised the original paragraph to clarify this point; please refer to the highlighted section in Lines 215 - 227.

● Regarding the sentence about the training architecture that lacked clarity:

We apologize for the confusion caused and have modified the sentence to ensure better clarity. The revised sentence now states: "We adhered to the table pre-training via execution [15] training architecture, which involves applying tabular question-answering downstream tasks with an end-to-end workflow. This begins with a BART base model pre-trained as a free-form language model to conduct the SQL synthetic. Subsequently, the pre-trained language model is fine-tuned using a multi-round tuning method with various datasets, including the Wikipedia Question Answering Dataset and the Metadata Question Answering Dataset."

● Regarding the low resolution of figures:

We have enhanced the resolution of all the figures, including Figure 1, to facilitate better readability.

● Concerning typos:

We have rectified the typos pointed out by you and also performed a comprehensive review of the manuscript to eliminate any remaining errors.

Reviewer #2:

● About the evaluation on a single dataset:

We acknowledge your concern regarding the robustness of our method. To this end, we have evaluated our model on an additional Table QA dataset. The results from this are included in the revised manuscript after Line 392. These results corroborate the effectiveness of our method on multiple datasets, thereby strengthening our claim of robustness.

● About the statistical strength of the experiments:

We concur that the inclusion of statistical values is fundamental to establishing the validity of our findings. We have now provided the average and standard deviation from 10 independent experiments with different random seeds in the revised manuscript. We have also specified the values of the random seeds in Lines 339 - 340.

● Regarding data and code availability:

We understand the importance of data and code accessibility for reproducibility. In line with the PLOS Data policy, we have made the Metadata Question Answering dataset privately available for peer review via Dryad, an official data repository for research publications. Here is the link: https://datadryad.org/stash/share/iBAzVXtzlu_-mBVXbKuAu66i8raXZZb1SYKtO84ReLE

Furthermore, we have updated the size of the dataset according to our funded project's increasing volume. However, as much of the code is used in our funder’s project, we may not be able to make the entire code publicly available immediately. We will continue to communicate with our funder and carry out Code Sanitization in all details of our Code and Data as the project and publication progress.

● On redundant content and minimal implementation details:

We have thoroughly revised the manuscript to remove any redundancies and have bolstered the depth of the implementation details to provide a more comprehensive understanding of our method. Specific revisions include:

In the abstract, we scrutinized the grammar and excised the superfluous description of the GCOOS interface. In Lines 1 - 9, we removed unnecessary information and streamlined our language to ensure readability and clarity.

In Lines 17 - 18, we eliminated the current search results introduction of the traditional method to avoid repetition.

In Lines 36 - 45, we simplified our discussion of the FAIR principle and reframed the introduction of the language model to more directly and effectively convey the central ideas.

Lines 81 - 89 saw the removal of redundant background information on the GCOOS platform. We believe this change allows the reader to focus more on the core substance of our research without being distracted by superfluous context.

In Lines 91 - 102, we overhauled the background of the traditional keyword-based search method to ensure that it more effectively complements and supports our method.

In Lines 116 - 125, we refined our description of relevant question-answering research to enhance the context and background against which our research is positioned.

Finally, in Lines 129 - 136, we reworked the discussion of the challenges of question-answering models to better highlight the issues our research seeks to address.

Additional modifications, indicated in the highlighted text within the manuscript, pertain to grammar corrections and typo fixes.

We hope that these modifications have improved the readability and clarity of our manuscript and look forward to your further feedback.

● Regarding the quality of figures and their placement:

We have enhanced the quality of the figures for improved readability and have integrated them within the manuscript to facilitate better comprehension.

● Concerning typos:

We have corrected the typos mentioned and conducted a thorough review of the manuscript to identify and rectify any remaining errors.

We trust that our revisions have satisfactorily addressed your concerns. We greatly value your time and effort in reviewing our manuscript. We are confident that these modifications have considerably enhanced the quality of our work and we look forward to receiving further feedback.

Sincerely,

Zhe Zhang

---

## [Decision Letter · Decision Letter 1]

4 Oct 2023

MetaQA: Enhancing Human-Centered Data Search using Generative Pre-trained Transformer (GPT) Language Model and Artificial Intelligence

PONE-D-23-10676R1

Dear Dr. Zhang,

We’re pleased to inform you that your manuscript has been judged scientifically suitable for publication and will be formally accepted for publication once it meets all outstanding technical requirements.

Kind regards,

Baby Gobin

Academic Editor

PLOS ONE

Additional Editor Comments (optional):

Reviewers' comments:

Reviewer's Responses to Questions

**Comments to the Author**

1. If the authors have adequately addressed your comments raised in a previous round of review and you feel that this manuscript is now acceptable for publication, you may indicate that here to bypass the “Comments to the Author” section, enter your conflict of interest statement in the “Confidential to Editor” section, and submit your "Accept" recommendation.

Reviewer #1: All comments have been addressed

Reviewer #2: All comments have been addressed

2. Is the manuscript technically sound, and do the data support the conclusions?

Reviewer #1: Yes

Reviewer #2: Yes

3. Has the statistical analysis been performed appropriately and rigorously? 

Reviewer #1: Yes

Reviewer #2: Yes

4. Have the authors made all data underlying the findings in their manuscript fully available?

Reviewer #1: Yes

Reviewer #2: Yes

5. Is the manuscript presented in an intelligible fashion and written in standard English?

Reviewer #1: Yes

Reviewer #2: Yes

6. Review Comments to the Author

Reviewer #1: The authors clarified satisfactorily the questions sent by reviewers. The paper improved in terms of the quality of writing and terms of scientific contribution. New experiments were added to the article. I think the paper can be accepted in its current form.

Reviewer #2: The authors addressed all the concerns raised in my previous comment: Results on an additional dataset, named WikiTableQuestions, are included in the revised version of the manuscript. Private link to the data is available to the reviewers, and the authors explained the availability issues and the plan. The authors added a statement about independent runs and the random seeds selection criteria.

7. PLOS authors have the option to publish the peer review history of their article (what does this mean?). If published, this will include your full peer review and any attached files.

Reviewer #1: No

Reviewer #2: No

---

## [Editor Report · Acceptance letter]

17 Oct 2023

PONE-D-23-10676R1 

MetaQA: Enhancing Human-Centered Data Search using Generative Pre-trained Transformer (GPT) Language Model and Artificial Intelligence 

Dear Dr. Zhang:

I'm pleased to inform you that your manuscript has been deemed suitable for publication in PLOS ONE. Congratulations! Your manuscript is now with our production department. 

Kind regards, 

on behalf of

Dr. Baby Gobin 

Academic Editor

PLOS ONE